# Enriching Lives: Geriatricians’ Mission of Supportive Care for Older Persons

**DOI:** 10.3390/geriatrics8060106

**Published:** 2023-10-26

**Authors:** Virginia Boccardi

**Affiliations:** Section of Gerontology and Geriatrics, Department of Medicine and Surgery, University of Perugia, Piazzale Gambuli 1, 06132 Perugia, Italy; virginia.boccardi@unipg.it; Tel.: +39-075-5783524

**Keywords:** aging, care, diseases, frailty, geriatrics, palliative

## Abstract

The growing older population, the increasing prevalence of chronic illnesses, and the pressing need to enhance the quality of end-of-life care have underscored the importance for geriatricians to focus on supportive and palliative measures. Within this context, the emphasis on delivering comprehensive and person-centered care has become crucial, ensuring that individuals not only receive medical treatment but also experience comfort during chronic illness and in their final days. Nevertheless, a significant number of older individuals often find themselves in hospitals during their last moments, sometimes undergoing aggressive medical interventions even when facing terminal conditions. The challenge lies in the early recognition of the end-of-life phase, initiating timely supportive and palliative care in conjunction with conventional treatments, adopting a multidisciplinary approach, and providing crucial support to grieving family members. Despite being a relatively recent field, geriatric palliative care (GPC) still requires further development. With this perspective, I aimed to shed light on the challenges and best practices for enhancing care for individuals facing chronic critical illnesses and frailty.

## 1. Introduction

In the past, the care of those who were at the end of their lives was deeply personal and was performed by the family. Loved ones provided comfort, herbal remedies and prayers. Although these gestures of compassion were heartfelt, they often fell short of addressing the profound physical and emotional distress that could accompany terminal illnesses. As civilizations evolved and medical knowledge advanced, the approach to death began to change. Scientific progression, the development of hospitals and the evolution of many professional figures have allowed for more specific treatment of diseases, often negatively affecting the comfort and dignity of individuals in their final days. “How people die remains in the memory of those who live on”, said Dame Cicely Saunders (Barnet, 22 June 1918–London, 14 July 2005), a British nurse, physician and philosopher of Christian faith (she converted from atheism to Anglicanism) [1]. Cicely Saunders recognized a profound need for a more holistic approach to care. She believed that it was not only essential to address the physical pain and symptoms of patients but also their emotional and spiritual suffering. In 1967, she established St. Christopher’s Hospice in London, a pioneering institution that would become the embodiment of her vision [1]. Here, patients received comprehensive care that encompassed pain management, psychological support and spiritual guidance. The approach was holistic, taking care of not only the length of life but also its quality. As time passed, Cicely’s innovative approach gained recognition and followers worldwide. The World Health Organization (WHO) recognized the importance of palliative care in the 20th century, declaring it an essential component of healthcare. This recognition led to further progress in the field, including the development of guidelines and the promotion of pain management as well as nutrition, supporting the importance of end-of-life discussions and care planning.

## 2. Where and How Do Older Individuals Pass Away Today?

A significant number of older persons spend their final moments in hospitals, often receiving aggressive medical treatments even in the face of terminal conditions [2]. The field and importance of end-of-life care for older adults has evolved significantly in the last years. For approximately a century and a half, many western countries witnessed a progressive shift in the location of where people pass away, moving from homes to hospitals. In older individuals in the terminal stages of illnesses, burdened by varying degrees of disability, conventional treatments are frequently administered despite negative prognostic factors and a clinical course that steadily worsens [3]. In such a context, the concept of dying has become less defined. In the past, an individual would be healthy, then fall ill and either recover or pass away swiftly. Only conditions like mental illness and tuberculosis consistently deviated from this pattern. However, this framework does not translate well to most chronically ill individuals. It assumes a stark transition in which patients move into a “dying” state by becoming “terminally ill”, necessitating a distinct form of care from patients who might either recover or maintain stability. Numerous chronically ill older individuals, such as subjects affected by dementia or heart failure [4], grapple with uncertain medical prognoses; they might be unwell enough to face mortality yet could also continue living for many years [5]. A more pragmatic perspective on this state of “near death” involves focusing on frailty rather than the exact timing of death. From this point of view, those living with serious illnesses at the end of their lives can be identified not by a certain prediction of when death will occur, but by existing in a state of “living on thin ice”—enduring extended periods of ailment or disability, reduced functionality and the potential for symptom exacerbation, any of which could prove fatal. This state could persist for several years, or a passing could transpire within a week. Patients in the terminal phase of a serious illness or even at the end of a prolonged disability resulting from multiple health issues may exhibit symptoms that are characteristic of the end-of-life stage for an extended period. These signs and indications go beyond just the sensation of discomfort. They encompass a range of experiences, including weariness, queasiness, retching, difficulty in breathing and mental confusion. As a result, a notable shift is underway, acknowledging the importance of embracing a broader and more person-centered strategy for tending to those in the latter phases of their lives. In this context, a movement to create Age-Friendly Health Systems is underway in the United States [6]. The objective is to establish healthcare systems that guarantee optimal care for older people, safeguarding them from any potential harm while ensuring their satisfaction with the care they receive. Employing the 4 Ms framework [6] places a distinct emphasis on addressing what Matters, Medication management, Mentation (cognitive health) and Mobility. When implemented, this framework has proven to be highly effective in achieving these goals. 

Collectively, there is a growing emphasis on affording older adults the opportunity to spend their remaining days in settings that foster ease, respect and an enhanced quality of existence. This has resulted in the emergence of alternative locations, including hospices, centers dedicated to palliative care and even care administered in one’s own residence. These settings prioritize shifting away from intensive medical interventions and instead focus on symptom management, pain relief and attending to the emotional and spiritual requirements of everyone. This change is driven by the understanding that maintaining a high quality of life in the final stages is often more valuable to the older person and their families than relentless medical interventions [7]. Through delivering all-encompassing support that addresses the older individual’s physical and emotional welfare, the objective is to facilitate a quiet and purposeful passage, encircled by close ones and within an atmosphere that aligns with their choices. In essence, the question of where the oldest pass away today is indicative of a broader shift in societal values, highlighting a movement toward personalized, compassionate and dignified end-of-life care that respects the individual’s wishes and optimizes their overall well-being during their final journey: what Matters. In such instances, the focus should shift toward prioritizing the patient’s quality of life (QoL) as the desired outcome, achieved through the application of palliative care (PC) [8]. 

## 3. What Does Palliative Care Entail?

The National Institute on Aging, according to Frequently Asked Questions About Palliative Care, National Institute on Aging (nih.gov) [9], defines palliative care as “specialized medial care for people living with serious illness. It can be received at the same time as treatment for the disease or condition and focuses on providing relief from the symptoms and stress of serious illness” [10]. The aim of such a discipline is to enhance the quality of life for patients and families struggling with incurable and terminal illnesses through the prevention and relief of sufferance through the early assessment and management of pain as well as other physical, psycho-social and spiritual issues. As specifically reported by the Palliative Care Advisory Council in 2018 [11], palliative care

Provides relief from pain and other distressing symptoms;Affirms life and regards dying as a normal process;Intends neither to hasten nor postpone death;Integrates the psychological and spiritual aspects of patient care;Offers a support system to help patients live as actively as possible until death;Offers a support system to help the family cope during the patient’s illness and in their own bereavement;Uses a team approach to address the needs of patients and their families, including bereavement counseling, if indicated;Enhances quality of life and may also positively influence the course of illness;Is applicable early in the course of illness in conjunction with other therapies that are intended to prolong life, such as chemotherapy or radiation therapy, and includes those investigations needed to better understand and manage distressing clinical complications.

According to the American Cancer Society, “palliative care” (or supportive care) is care that focuses on relieving symptoms caused by serious illnesses like cancer. It can be performed at any time during a person’s illness to help them feel more comfortable. The terms “supportive care” and “palliative care” are now often used interchangeably. However, supportive care, as defined by the National Cancer Institute, refers to care given to improve the quality of life of patients who have a serious or life-threatening disease [12]. The overarching aim of supportive care revolves around mitigating or addressing the symptoms associated with a disease, managing the adverse effects of treatment and addressing the psychological, social and spiritual facets intertwined with the disease or its treatment. This approach is commonly recognized by various names, including comfort care, palliative care and symptom management. Both palliative care and supportive care are united in their central focus on patients and their families, aiming to enhance or safeguard their quality of life. However, a subtle distinction emerges. Whereas endeavors aimed at alleviating pain and distress associated with cancer find wide acceptance under the umbrella of “supportive care”, the term “palliative care” sometimes instigates anxiety among patients, families and healthcare providers, owing to a misconception that it implies a relinquishment of treatment endeavors. This enduring misperception, which permeates not only among patients and their families but also within the healthcare community, equating “palliative care” with “end-of-life care”, poses a significant barrier to patients and families accessing the crucial services they require. Consequently, some patients and providers perceive “supportive care” as a more neutrally expressed alternative for the same services, potentially diminishing some of the emotional resistance linked to seeking care beyond curative treatments.

## 4. Where Is the Challenge of Palliative Care in Geriatric Medicine?

Geriatric medicine, a medical subspecialty, addresses the unique healthcare needs of the older and frail population. Frailty is theoretically defined as a clinically recognizable state of increased vulnerability resulting from aging-associated decline in reserve and function across multiple physiologic systems, such that the ability to cope with everyday or acute stressors is compromised [13]. Although geriatric medicine provides comprehensive care for older adults in general, its specialized focus caters to the needs of particularly older patients. Notably, the prevalence of conditions best addressed by geriatric medicine significantly rises in individuals aged 80 and above. This group often exhibits high levels of frailty and contends with multiple ongoing chronic health problems [13].

Geriatric medicine relies on multidimensional and interdisciplinary assessments, making it a meta-discipline within the medical field [14]. The challenge in this field lies in the early identification of the so-called end-of-life phase, initiating timely PC alongside traditional treatments, employing a multidisciplinary approach and providing valuable support to family members during the grieving process. Epidemiological studies show that, on average, the final two decades of an individual’s life in Italy are marked by a progressive accumulation of chronic multimorbidity, functional reliance, frailty and, frequently, cognitive decline [15]. Concurrently, as the causes of death undergo a transformation, the dynamics of the dying phase evolve, and the final stages of life stretch into a prolonged period marked by difficult treatment decisions, challenging symptom control, multifaceted psychosocial issues and often unaddressed spiritual anguish. Hence, the imperative for hospice and palliative care tailored to the distinctive needs and circumstances of the older person becomes unmistakably clear, particularly given the rising population residing in assisted living facilities and residential care homes. Palliative care should be introduced gradually before clinical challenges become overwhelming. Teaching the art of end-of-life conversations contributes to realizing patient-centered medicine. The choice not to limit palliative care, as often still happens, to the so-called “last days’ care”, usually not exceeding two weeks, requires synergy among the family physician, oncologist and palliative care specialist. Many patients with advanced illnesses struggle to enforce their potential wishes or advanced decisions, not only regarding cardiopulmonary resuscitation but also concerning any medical procedure and the overall treatment goals or other intensive therapy decisions, such as the use of artificial nutrition and hydration, as well as forced ventilation. In practice, communication with the patient about end-of-life concerns, care planning, the patient’s preferences, personal values and quality of life is essential. This discussion underscores various unique challenges associated with palliative care for individuals with dementia: delayed diagnoses, the complexity of predicting symptom progression, difficulty faced by individuals in articulating their symptoms and care preferences, and heightened sensitivity to medication side effects [16]. Expanding upon these foundational components, geriatric palliative care (GPC) can be characterized as an approach with the overarching objective of enhancing the well-being of older individuals who are confronting severe and potentially life-threatening illnesses during the final stages of their lives. Unlike geriatrics, which is delineated by the age group it serves, and palliative care, which is defined by its specific care objectives, GPC occupies a distinct position. It does not establish itself as an entirely new medical specialty or a subfield within either of these domains. Instead, GPC represents a collaborative effort that operates at the intersection of geriatrics and palliative care, fostering inter-specialty cooperation to address the multifaceted needs of older persons in challenging health circumstances. Despite its recent inception, GPC is still lacking evidence. Geriatrics and palliative care represent separate yet interconnected medical disciplines [17,18]. Both fields are characterized by their highly diverse and interdisciplinary nature, emphasizing patient- and family-centered approaches geared toward enhancing quality of life, individual capabilities and social involvement. The synergistic potential arising from the convergence of these closely related specialties can serve as a valuable model for fostering collaboration among different branches of healthcare [19]. Although integrated care and models ensuring continuity of care are critical at the healthcare provider level, it is equally important to promote closer collaboration among professional specialties, such as geriatrics and palliative care [17]. It is of great importance for professionals across various medical specialties to be well informed and proactive regarding the integration of early palliative and comfort care interventions for older adults. This imperative becomes even more significant given that, today, older individuals do not have the benefit of being always under the care of a geriatric specialist. Therefore, it is recommended that any healthcare provider, regardless of their specialty, takes the initiative to incorporate comfort care measures in collaboration with palliative care specialists. In situations involving frail and medically complex older adults, it is also prudent to consider referring them to geriatric specialists to ensure a seamless continuation of this essential partnership.

## 5. How Do You Decide on Appropriate Management at the End of Life?

Giovanni Paolo II in the Evangelium Vitae [20] wrote, “Therapeutic obstinacy refers to certain medical interventions that are no longer appropriate given the patient’s actual condition, either because they are now disproportionate to the expected outcomes or because they are too burdensome for the patient and their family. In these situations, when death is imminent and inevitable, it is ethically permissible to forego treatments that would only prolong a precarious and painful life, while not discontinuing the ordinary care due to the patient in such cases. There is certainly an ethic obligation to ensure medical care and treatment, but the intensity must be weighed against the specific cases and circumstances. It is necessary to assess whether the available therapeutic means are objectively proportionate to the prospects of improvement”.

The first paragraph of Article 16 [21], again, of the code of ethics reads as follows: “The physician, taking into account the patient’s expressed wishes or their legal representative’s, and considering the principles of effectiveness and appropriateness of care, does not undertake or persist in clinically inappropriate and ethically disproportionate diagnostic procedures and therapeutic interventions, from which an actual health benefit and/or an improvement in quality of life cannot reasonably be expected”. Thus, it appears that the physician should cease treatment only when it is unavoidable and when the patient’s biological end of life is imminent. Outside these scenarios, the physician is obligated to provide necessary care, and only the patient’s refusal of such care (also expressed in Advance Healthcare Directives, if prepared) then imposes upon the professional the duty not to continue medical treatment. However, exercising fundamental rights in managing one’s life is challenging and complex to realize, especially among the oldest old. Older adults, when hindered by cognitive and physical decline, are usually unable to express their wishes; years lived in pain and suffering then become a crucial judgment parameter. Implementing palliative care in the field requires experience; attention must be given to the patient’s and family’s environmental comfort and preferences, which are certainly influenced by the availability of specific services, economic capabilities and traditions. The challenge in geriatric medicine is intricately woven into the delicate balance between therapeutic perseverance and abandonment, all while grappling with the difficulty of identifying boundaries and acknowledging terminality. This complex terrain often navigates the fine line between providing medical interventions that could prolong life but might entail undue suffering and refraining from interventions that might hasten the inevitable, but it could also deny the individual a dignified and comfortable end. It is a matter of discerning when medical interventions become disproportionate to the potential benefits and when respecting the person’s natural course of life becomes paramount.

## 6. What about the Significance of End-of-Life Culture for the Geriatrician?

As geriatric medicine continues to evolve in response to the aging population, the role of the geriatrician in shaping end-of-life care becomes increasingly pivotal [22]. Although medical advancements enhance our ability to care for complex health conditions in frail persons, the importance of promoting an end-of-life culture in this field is extremely important. In essence, an end-of-life culture within geriatric medicine entails a comprehensive approach that goes beyond the technical aspects of medical care. It encompasses the values, beliefs and preferences of both patients and their families, aiming to provide a holistic and dignified transition in the twilight years. The geriatrician, as a primary provider for older adults, possesses a unique vantage point to influence this culture. One key aspect of fostering an end-of-life culture is facilitating open and compassionate conversations about the inevitable stages of life [23]. By engaging patients in discussions about their goals, preferences and fears, geriatricians can ensure that medical decisions align with the individual’s values. Such discussions not only empower patients to make informed choices but also grant families a roadmap for decision making in times of crisis. Furthermore, geriatric specialists have the potential to play a crucial role in dispelling the misconception that palliative care means giving up on treatment altogether. By articulating the principal importance of life quality as opposed to the mere extension of life, they can aid patients and their families in comprehending that end-of-life care does not signify relinquishing hope. Instead, it is about guaranteeing that the remaining time is marked by solace, honor and meaningful moments. Educational initiatives also constitute the foundation of cultivating a culture that embraces end-of-life considerations. Geriatric specialists can wield influence over their peers, budding medical practitioners and fellow healthcare providers by underscoring the significance of all-encompassing end-of-life care during their training. By advocating for interdisciplinary collaboration, they can establish a framework that acknowledges the psychological, social and spiritual dimensions of patients’ journeys, thereby reinforcing the centrality of a person-centric approach [7].

## 7. What about the Significance of Comprehensive Geriatric Assessment?

The significance of end-of-life culture for the geriatrician is profound. It underscores the crucial role these professionals play in shaping the twilight years of their patients with empathy, understanding and expertise. To foster a society that embraces extended lifespans, it is essential to establish integrated palliative care models specifically designed to cater to older individuals who have often been marginalized in the past [24]. Timely integration of palliative and supportive services holds significance in the comprehensive care of older and frail patients, irrespective of their disease stage. By fostering open dialogues, challenging misconceptions and championing holistic care, geriatricians have the power to transform the way society perceives and approaches end-of-life issues, ensuring that older and frail individuals experience a transition that is not only medically managed but emotionally supported and respectful of their wishes [25]. Considering this intricacy, it becomes evident that effective palliative and supportive care for older patients necessitates a comprehensive assessment encompassing various domains of their health. This assessment is often carried out through a comprehensive geriatric assessment (CGA), utilizing validated tools to gauge specific geriatric domains that hold significant relevance in the care this population [26]. The CGA encompasses an evaluation of comorbidities, a screening for cognitive impairment and delirium, an assessment of polypharmacotherapy, a determination of functional status, an assessment of physical performance (mobility), an evaluation of nutritional status, a scrutiny of falls and an examination for social issues, such as environmental factors, available resources and social support. Thus, interventions guided by CGA can prove particularly beneficial in tailoring personalized care plans for older adults undergoing palliative and supportive care. In addition, it is imperative for researchers to actively involve older patients and their caregivers in investigations pertaining to innovative palliative care interventions. This collaborative approach not only expands the body of evidence but also fosters the development of customized care delivery approaches. It is essential that the realm of geriatric palliative care is firmly integrated into the healthcare system, seamlessly blending with cancer care to enhance the quality of services offered to an increasingly vulnerable and expanding demographic. Each person should have the chance to anticipate a gratifying existence as they advance in years. The realization of a longevous society may remain elusive without fair and accessible palliative care for all.

## 8. What Lies Ahead for Strategies in Geriatric Palliative Care?

The future direction for geriatric palliative care is likely to focus on several key areas to address the evolving needs of older adults including personalized care plans, interdisciplinary teams, early integration, education and training, research, and evidence-based practices and support for caregivers. A tailored palliative care approach to the individual needs, preferences and goals of each older patient looks essential. This involves comprehensive assessments, as mentioned earlier, to create customized care plans. Promoting collaboration among healthcare experts spanning diverse domains, including geriatrics, oncology, psychology and social work, becomes imperative. This collaborative effort aims to deliver all-encompassing care, encompassing the facets of physical, emotional and social dimensions that accompany the aging process and illness. The call to action here is to underscore the pressing requirement to acknowledge the significance of integrating palliative care at an early juncture. This integration should not merely be reserved for advanced stages but ought to commence from the very moment of diagnosis or even earlier. This can improve outcomes and reduce unnecessary aggressive treatments. Providing specialized training for healthcare professionals to better understand and address the unique needs and challenges of older adults receiving palliative care is also mandatory.

Furthermore, there is a compelling need to engage in research endeavors aimed at broadening the foundation of evidence supporting interventions in the realm of geriatric palliative care. Such efforts can substantially enhance the caliber of care bestowed upon individuals. And last, but by no means least, it stands as imperative to acknowledge and cater to the requirements of family caregivers, who assume pivotal roles in the care of older adults grappling with severe illnesses. Furnishing them with the necessary resources and bolstering support mechanisms becomes essential.

In memory of my mother-in-law, Maria Camilla, who passed away at the age of 69 after 14 years of battling Alzheimer’s Disease.

## Data Availability

No data has been used for this perspective.

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
