# Peer review of "Enriching Lives: Geriatricians’ Mission of Supportive Care for Older Persons"

_geriatrics, 2023, doi:10.3390/geriatrics8060106_

Round 1
Reviewer 1 Report
Comments and Suggestions for Authors
The manuscript addresses a relevant theme that is little discussed in studies on human aging, which is the performance of geriatric professionals. The text is well written, with an excellent theoretical foundation and updated references.
I suggest that paragraphs 2 to 10 be divided, because they are quite extensive which can hinder the idea addressed.
Author Response
I express my sincere gratitude to the Reviewer for the valuable suggestions and feedback, which have greatly contributed to the revision and improvement of our manuscript. The input has been important in refining our study and strengthening its overall scientific merit.
Reviewer #1: The manuscript addresses a relevant theme that is little discussed in studies on human aging, which is the performance of geriatric professionals. The text is well written, with an excellent theoretical foundation and updated references.
I suggest that paragraphs 2 to 10 be divided, because they are quite extensive which can hinder the idea addressed.
Response: I want to express my sincere gratitude for your invaluable feedback. As per your kind suggestion, I have divided the text into paragraphs. Your input has been greatly appreciated.
Reviewer 2 Report
Comments and Suggestions for Authors
Thank you for submission of this important manuscript discussing the need to incorporate palliative care and geriatrics to provide the highest quality care for older adults living with chronic comorbidities.
1. While the manuscript focuses on collaborative care between geriatrics and palliative care, it is critical for all specialties to be aware of and consider early palliative/comfort care interventions for older adults. This is especially important as most older adults are not cared for by a geriatric specialist. I would suggest a sentence about this, highlighting the importance of any provider initiating comfort care measure in partnership with a palliative care specialist. You could also note that a referral to geriatrics to continue this partnership could be considered in cases of frail, medically complex older adults.
2. I would suggest not using the cited WHO definition of palliative care. Pains are taken in the manuscript to highlight the importance of implementing palliative care early in the course of a life-threatening disease and it is noted that palliative care can be continued in conjunction with life-prolonging therapy. I am concerned that the wording of the definition used from the WHO is in disagreement with the accepted role of palliative care as an important part of care that can be started early in a serious illness and continued along with other treatments. I would suggest use of a definition that better encompasses these concepts, such as the one from the National Institute on Aging at Frequently Asked Questions About Palliative Care | National Institute on Aging (nih.gov). It defines palliative care as “specialized medial care for people living with serious illness. It can be received at the same time as treatment for the disease or condition and focuses on providing relief from the symptoms and stress of serious illness”.
3. The manuscript references engaging with older adults and their families around what brings quality to their life/values at the end of life. I would suggest referencing the Age-Friendly Health Systems 4 Ms framework, with a focus on What Matters. This system focuses on moving away from a problem-centric view of medicine to focus care around What Matters to older adults. This could even be expanded to briefly reference Patient Priorities Care, one of the preferred frameworks for eliciting and acting on What Matters. This is often helpful to clinicians as a structured process to move from What Matters to thinking about how the older adult’s goals identified health priorities will impact the plan of care.
Comments on the Quality of English Language
1. The English of the manuscript needs review. I note several miss-spellings/typos and the punctuation is not correct in several places. This is primarily an issue with the placement of commas. I would suggest review by a native English speaker if possible.
2. In the United States, the term “older adults” has replaced many other terms for individuals of advance age including elderly, seniors, and senior citizens. I would recommend you make this substitution as the other terms are felt to have connotations of ageism.
Author Response
I express my sincere gratitude to the Reviewer for the valuable suggestions and feedback, which have greatly contributed to the revision and improvement of our manuscript. The input has been important in refining our study and strengthening its overall scientific merit.
Reviewer #2:
Thank you for submission of this important manuscript discussing the need to incorporate palliative care and geriatrics to provide the highest quality care for older adults living with chronic comorbidities.
- While the manuscript focuses on collaborative care between geriatrics and palliative care, it is critical for all specialties to be aware of and consider early palliative/comfort care interventions for older adults. This is especially important as most older adults are not cared for by a geriatric specialist. I would suggest a sentence about this, highlighting the importance of any provider initiating comfort care measure in partnership with a palliative care specialist. You could also note that a referral to geriatrics to continue this partnership could be considered in cases of frail, medically complex older adults.
Response: I greatly appreciate your feedback. Kindly note that this concept has been introduced at the end of paragraph 4. Thank you for your time and consideration.
- I would suggest not using the cited WHO definition of palliative care. Pains are taken in the manuscript to highlight the importance of implementing palliative care early in the course of a life-threatening disease and it is noted that palliative care can be continued in conjunction with life-prolonging therapy. I am concerned that the wording of the definition used from the WHO is in disagreement with the accepted role of palliative care as an important part of care that can be started early in a serious illness and continued along with other treatments. I would suggest use of a definition that better encompasses these concepts, such as the one from the National Institute on Aging at Frequently Asked Questions About Palliative Care | National Institute on Aging (nih.gov). It defines palliative care as “specialized medial care for people living with serious illness. It can be received at the same time as treatment for the disease or condition and focuses on providing relief from the symptoms and stress of serious illness”.
Response: Thank you for the suggestion; the sentence has been changed accordingly.
- The manuscript references engaging with older adults and their families around what brings quality to their life/values at the end of life. I would suggest referencing the Age-Friendly Health Systems 4 Ms framework, with a focus on What Matters. This system focuses on moving away from a problem-centric view of medicine to focus care around What Matters to older adults. This could even be expanded to briefly reference Patient Priorities Care, one of the preferred frameworks for eliciting and acting on What Matters. This is often helpful to clinicians as a structured process to move from What Matters to thinking about how the older adult’s goals identified health priorities will impact the plan of care.
Response: I greatly appreciate your feedback. Kindly note that this concept has been added in the revised manuscript.
Comments on the Quality of English Language
- The English of the manuscript needs review. I note several miss-spellings/typos and the punctuation is not correct in several places. This is primarily an issue with the placement of commas. I would suggest review by a native English speaker if possible.
Response: Thank you for the revisions. The manuscript has been enhanced in terms of English spelling and grammar. It's worth noting that MDPI also provides a service for pre-publication checks.
- In the United States, the term “older adults” has replaced many other terms for individuals of advance age including elderly, seniors, and senior citizens. I would recommend you make this substitution as the other terms are felt to have connotations of ageism.
Response: Thank you so much. I have made all the necessary substitutions.
Reviewer 3 Report
Comments and Suggestions for Authors
Dear Editor,
Thank you for the opportunity to review this manuscript. In this manuscript, the author proposed geriatricians’ missions of supportive care for seniors. Ideas are conveyed in the well manner, starting from describing the situations of passing away nowadays and ending with strategies in geriatric palliative care.
While the title is interesting and might fit with the journal, some parts of the manuscript need to be improved as below:
1. There are several grammatical and writing errors, for example:
- Page 1 line 27 “olistic”, do you mean “holistic”?
- Page 3 line 105 “perfromed”, do you mean “performed”?
- Page 3 line 143, there is no full stop between two sentences.
- Page 4 line 160, there is no space between full stop and new sentence.
- Page 5 line 232-245 is written with the different font size.
- Please check throughout the manuscript. There are other writing errors as well.
2. Some sentences definitely need citation, but the authors did not provide citation, for example:
- Page 4 line 183-193. It is written “Giovanni Paolo II in the EVANGELIUM VITAE wrote: "Therapeutic obstinacy refers to certain medical interventions … (and so on).” While it is clear that this statement belongs to Giovanni Paolo II, authors did not provide any source until the end of the paragraph.
- Page 4 line 193-198, it is written “Article 16, first paragraph, of the code of ethics reads as follows: "The physician, taking into account the patient's expressed wishes or their legal representative's, … (and so on).” While this statement clearly belongs to code of ethics, citation is not provided for this statement.
Author Response
Reviewer #3:
Dear Editor,
Thank you for the opportunity to review this manuscript. In this manuscript, the author proposed geriatricians’ missions of supportive care for seniors. Ideas are conveyed in the well manner, starting from describing the situations of passing away nowadays and ending with strategies in geriatric palliative care.
While the title is interesting and might fit with the journal, some parts of the manuscript need to be improved as below:
- There are several grammatical and writing errors, for example:
- Page 1 line 27 “olistic”, do you mean “holistic”?
- Page 3 line 105 “perfromed”, do you mean “performed”?
- Page 3 line 143, there is no full stop between two sentences.
- Page 4 line 160, there is no space between full stop and new sentence.
- Page 5 line 232-245 is written with the different font size.
- Please check throughout the manuscript. There are other writing errors as well.
Response: Thank you so much. I have made all the necessary substitutions. The manuscript has been also enhanced in terms of English spelling and grammar. It's worth noting that MDPI also provides a service for pre-publication checks.
- Some sentences definitely need citation, but the authors did not provide citation, for example:
- Page 4 line 183-193. It is written “Giovanni Paolo II in the EVANGELIUM VITAE wrote: "Therapeutic obstinacy refers to certain medical interventions … (and so on).” While it is clear that this statement belongs to Giovanni Paolo II, authors did not provide any source until the end of the paragraph.
- Page 4 line 193-198, it is written “Article 16, first paragraph, of the code of ethics reads as follows: "The physician, taking into account the patient's expressed wishes or their legal representative's, … (and so on).” While this statement clearly belongs to code of ethics, citation is not provided for this statement.
Response: Thank you so much. The references have been properly added.
Round 2
Reviewer 1 Report
Comments and Suggestions for Authors
After reevaluating the article, I issue a favorable opinion on its publication, as the requests for adjustments were met.
Reviewer 2 Report
Comments and Suggestions for Authors
Following your revisions and response to my prior comments, I have no further concerns and recommend this manuscript for publication.
Reviewer 3 Report
Comments and Suggestions for Authors
Dear Editor,
Thank you for the opportunity to review the revised manuscript file. I appreciate author's efforts to revise this manuscript. All comments were addressed very well. I trust this manuscript can be accepted with this present form. Thank you.